# Reactive Oxygen Species in the Aorta and Perivascular Adipose Tissue Precedes Endothelial Dysfunction in the Aorta of Mice with a High-Fat High-Sucrose Diet and Additional Factors

**DOI:** 10.3390/ijms24076486

**Published:** 2023-03-30

**Authors:** Ayumu Osaki, Kazuki Kagami, Yuki Ishinoda, Atsushi Sato, Toyokazu Kimura, Shunpei Horii, Kei Ito, Takumi Toya, Yasuo Ido, Takayuki Namba, Nobuyuki Masaki, Yuji Nagatomo, Takeshi Adachi

**Affiliations:** Department of Internal Medicine I, Division of Cardiovascular Medicine, National Defense Medical College, 3-2 Namiki, Tokorozawa 359-8513, Japan

**Keywords:** type 2 diabetes mellitus, steatohepatitis, hypercholesterolemia, perivascular adipose tissue, endothelial dysfunction, oxidative stress

## Abstract

Metabolic syndrome (Mets) is the major contributor to the onset of metabolic complications, such as hypertension, type 2 diabetes mellitus (DM), dyslipidemia, and non-alcoholic fatty liver disease, resulting in cardiovascular diseases. C57BL/6 mice on a high-fat and high-sucrose diet (HFHSD) are a well-established model of Mets but have minor endothelial dysfunction in isolated aortas without perivascular adipose tissue (PVAT). The purpose of this study was to evaluate the effects of additional factors such as DM, dyslipidemia, and steatohepatitis on endothelial dysfunction in aortas without PVAT. Here, we employed eight-week-old male C57BL/6 mice fed with a normal diet (ND), HFHSD, steatohepatitis choline-deficient HFHSD (HFHSD-SH), and HFHSD containing 1% cholesterol and 0.1% deoxycholic acid (HFHSD-Chol) for 16 weeks. At week 20, some HFHSD-fed mice were treated with streptozocin to develop diabetes (HFHSD-DM). In PVAT-free aortas, the endothelial-dependent relaxation (EDR) did not differ between ND and HFHSD (*p* = 0.25), but in aortas with PVAT, the EDR of HFHSD-fed mice was impaired compared with ND-fed mice (*p* = 0.005). HFHSD-DM, HFHSD-SH, and HFHSD-Chol impaired the EDR in aortas without PVAT (*p* < 0.001, *p* = 0.019, and *p* = 0.009 vs. ND, respectively). Furthermore, tempol rescued the EDR in those models. In the Mets model, the EDR is compromised by PVAT, but with the addition of DM, dyslipidemia, and SH, the vessels themselves may result in impaired EDR.

## 1. Introduction

Obesity is a growing healthcare problem worldwide, with increasing prevalence, high mortality, and hospitalization rates, and high medical costs. Diet-induced obesity (including diets high in fat, sugar, and salt) and metabolic syndrome (Mets) are the major contributors to the onset of metabolic complications, such as hypertension, type 2 diabetes mellitus (T2DM), and dyslipidemia, resulting in cardiovascular diseases (CVDs), cancer, and non-alcoholic fatty liver disease (NAFLD) [1].

In order to understand the pathophysiology of Mets, a rodent model fed a high-fat and high-sucrose diet (HFHSD) has been used [2,3,4,5]. The contribution of HFHSD to insulin resistance depends on the strains of mice, and C57BL/6 mice were demonstrated to develop insulin resistance by HFHSD [6]. Insulin is known to cause not only hypoglycemic but also endothelial cell-dependent vasorelaxation, which is thought to be due to its mechanism of promoting nitric oxide (NO) biosynthesis via the insulin receptor substrate (IRS)/phosphatidylinositol-3-kinase (PI3K)/Akt pathway [7,8]. In Mets with insulin resistance, this IRS/PI3K/Akt/endothelial nitric oxide synthase (eNOS)/NO pathway is impaired, and the endothelial cell-dependent vasorelaxation is attenuated [9,10]. However, Huige’s group and our group found that an HFHSD with insulin resistance induced minimal endothelial dysfunction in mouse aortas [5]. Huige revealed that an HFHSD impaired endothelial dysfunction in the aortas with perivascular adipose tissue (PVAT) due to eNOS uncoupling in PVAT [11,12]. We hypothesized that in an HFHSD, the perivascular component alone is impaired, and the additional factors impair not only PVAT but also the vessels themselves. We considered T2DM, NAFLD, and hypercholesterolemia as additional factors. T2DM is associated with endothelial dysfunction, which accelerates atherosclerosis, resulting in CVDs [13]. Many obese patients in Mets have NAFLD, which is associated with insulin retorrence [14,15,16]. Patients with NAFLD have a significant increase in the risk of CVDs [17]. Hypercholesterolemia is also associated with endothelial dysfunction [18]. It has also been reported that under a chronic HFHSD, PVAT shows increased macrophage infiltration and inflammatory cytokine expression, which promotes atherosclerosis [19].

In this study, we created HFHSD-based mouse models of T2DM, NAFLD, and hypercholesterolemia to investigate the factors of endothelial dysfunction that occur in addition to insulin resistance, including PVAT.

## 2. Results

### 2.1. Changes in Body and Organ Weights in Each Group

All the HFHSD groups gained more body weight than the ND group (Table 1). The body weight of the HFHSD-SH and HFHSD-DM groups modestly decreased compared with the HFHSD group (Table 1 and Appendix A). The liver weights in the HFHSD-Chol group were the most enlarged among all the groups (Table 1). We also measured the weights of adipose tissue (GWAT, SWAT, and PVAT). In all HFHSD groups, the weights of GWAT were increased compared with the ND group. The HFHSD-Chol group gained more GWAT weight than the other HFHSD groups. SWAT and PVAT weights were increased in all HFHSD groups, but those of the HFHSD-DM and HFHSD-SH groups were modestly decreased compared with the HFHSD and HFHFD-Chol groups (Table 1).

### 2.2. Glucose Tolerance and Insulin Sensitivity Is Impaired in HFHSD Groups, and HFHSD-DM Group Shows Impaired Insulin Secretion

We confirmed that all HFHFD-fed groups had increased liver and adipose tissue weights. These are major organs in insulin resistance, and as such, we assessed insulin levels and insulin resistance. All HFHFD-fed groups had increased fasting glucose levels compared to the ND group. The HFHSD-DM group in particular showed the highest glucose level (Table 2). Insulin levels also decreased in the HFHSD-DM group following the injection of STZ (Table 2). Because the HFHSD markedly aggravated insulin resistance, we assessed OGTT and ITT in order to evaluate glucose tolerance and insulin sensitivity. All HFHFD-fed groups had impaired glucose tolerance compared with the ND group in OGTT, and the HFHSD-DM group in particular showed the highest levels of glucose (Figure 1A–D). All HFHFD-fed groups also showed insulin resistance in the ITT (Figure 1E–H). In particular, the HFHSD-Chol group showed the highest insulin resistance (Figure 1H).

### 2.3. HFHSD-SH Group Had the Worst Hepatosteatitis and the HFHSD-Chol Group Had the Worst Steatohepatitis

We showed that all HFHSD-fed groups had aggravated fasting glucose levels, glucose tolerance, and insulin sensitivity. Fasting glucose is mainly responsible for liver function, and steatohepatitis is closely related to insulin resistance. Therefore, we assessed steatohepatitis in each group. All HFHSD-fed groups showed many lipid drops in the liver with hematoxylin and eosin staining (Figure 2A) and oil red O staining (Figure 2B). In all HFHSD-fed groups, serum ALT levels were increased compared with the ND group, and the HFHSD-Chol group in particular had increased ALT levels compared with other HFHSD groups (Figure 2D). We also assessed hepatic TG and cholesterol. Hepatic TG was increased in all HFHSD-fed groups compared with the ND group (Figure 2E). The hepatic cholesterol levels were also increased in the HFHSD-fed groups, excluding HFHSD-SH, and the HFHSD-Chol group was the most increased in hepatic cholesterol levels (Figure 2F). We scored the staging of NAFLD and fibrosis with NAFLD activity score and staging for NAFLD [20]. The steatohepatitis worsened in the HFHSD-Chol group, as evidenced by the NAFLD activity score (Figure 2G). The HFHSD-SH group in particular showed marked fibrosis in the liver, as evidenced by Masson’s trichrome staining (Figure 2H). These data showed that the HFHSD-SH group had the worst steatohepatitis, and the HFHSD-Chol group had the worst steatohepatitis among the five groups.

### 2.4. HFHSD-Chol Group Gained the Highest Levels of Serum Cholesterol

All HFHSD-fed groups revealed higher serum cholesterol levels than the ND group. The HFHSD-Chol group showed a higher serum cholesterol level than the other HFHSD-fed group (Table 2). The serum HDL-cholesterol was also increased in the HFHSD-fed groups, except in the HFHSD-DM group (Table 2). There was no difference in serum TG levels among the five groups (Table 2). These data showed that the HFHSD-Chol group in particular had the highest serum cholesterol levels.

### 2.5. HFHSD Groups Had a Decrease in Adiponectin and a Release of Inflammatory Adipokines

The adipose tissue secreted various hormones and cytokines (adipokines) to maintain metabolic homeostasis. The adipokines have been linked to hypertension, DM, and Mets. We measured some adipokines, such as adiponectin, leptin, and TNF-α, in order to assess the changes in adipose tissue because the weight of adipose tissue was increased in HFHSD groups. The serum adiponectin levels were decreased by HFHSD load, but there was no significant difference among the HFHSD groups (Table 2). Serum leptin and TNF-α levels were increased by the HFHSD load. The serum leptin levels were decreased in HFHSD-DM compared with the HFHSD group, but there was no significant difference in adiponectin and TNF-α among the four HFHSD groups (Table 2). These data showed that adipose tissues were also impaired in HFHSD loads, leading to a decrease in adiponectin and a release of inflammatory adipokines.

### 2.6. Systolic Blood Pressure and Heart Rate

Insulin resistance, lipid overflow, and inflammatory adipokines are closely associated with hypertension and endothelial dysfunction, which are the basis of CVDs. Then, we assessed the systolic blood pressure in each group. All HFHSD-fed groups revealed higher systolic blood pressure levels than the ND group, although the heart rate did not differ among the five groups (Appendix A).

### 2.7. HFHSD-DM, HFHSD-SH, and HFHSD-Chol Groups Showed Increased Superoxide Production and Endothelial Dysfunction

Endothelial dysfunction can be caused by vascular superoxide production. Arteriosclerosis, which causes hypertension, is caused by vascular endothelial damage. We assessed the vascular superoxide production because vascular oxidative stress has been observed to aggravate endothelial dysfunction. We assessed the superoxide production of aortic rings via the fluorescence of dihydroethidium (DHE) staining, which indicates superoxide production. Fluorescence in DHE staining was higher in the HFHSD-DM, HFHSD-SH, and HFHSD-Chol groups than the ND and HFHFD groups (Figure 3A,C) and was blunted with tempol (Figure 3B,C). To test the contribution of vascular superoxide production to endothelial function, we assessed the endothelial-dependent relaxation (EDR) in mice by examining the acetylcholine-induced relaxation in aortic rings without PVAT. The EDR showed no difference between the ND and HFHSD groups (Figure 3D). However, the EDR in the HFHSD-DM, HFHSD-SH, and HFHSD-Chol groups was significantly impaired compared with the EDR in the ND group (Figure 3E–G). Tempol restored the EDR to levels similar to the ND group (Figure 3E–G). These data suggested that the endothelial dysfunction of the HFHSD-DM, HFHSD-SH, and HFHSD-Chol groups could be caused by superoxide produced by the aortic vessels themselves.

### 2.8. PVAT Increased EDR in the ND Group but Decreased EDR in the HFHSD Group and Was Accompanied by Increased Superoxide Production from PVAT

Endothelial dysfunction is mainly caused by inflammation from vascular endothelial cells. It is assumed that inflammation, which promotes arteriosclerosis, spreads from the outside to the inside of the vessel. Most vessels are covered with PVAT. In the HFHSD groups, the weights of PVAT were higher than those in the ND group (Table 1), and some adipokine levels of the HFHSD-fed group changed compared with those of the ND group (Table 2). As such, we assessed the adipose superoxide production of the aortas with PVAT via DHE staining. Fluorescence in DHE staining was significantly increased in the aortas with PVAT from the HFHSD, HFHSD-DM, HFHSD-SH, and HFHSD-Chol groups (Figure 4A,C). These increases were restored by tempol to similar levels of the ND group (Figure 4B,C). Next, we assessed the EDR of the aortas with PVAT to evaluate the contribution of adipose superoxide production from PVAT to endothelial dysfunction. In the ND group, the EDR of the aortas with PVAT was increased compared with that of the aortas without PVAT. In contrast, in the HFHSD group, the EDR of the aortas with PVAT was decreased compared with that of the aortas without PVAT (Figure 4D). In the HFHSD-DM, HFHSD-SH, and HFHSD-Chol groups, the EDR of the aortas with PVAT showed no significant difference compared with that of the aortas without PVAT (Figure 4E–G). These data showed that PVAT increased the EDR in the ND group but decreased the EDR in the HFHSD group and was accompanied by increased superoxide production from PVAT.

## 3. Discussion

This study revealed two things. The first is that only the HFHSD minimally induced endothelial dysfunction, but additional factors, including hyperglycemia, steatohepatitis, and high cholesterol, induced endothelial dysfunction. Combining the HFHSD with injections of STZ induced sustainable hyperglycemia (HFHSD-DM). The choline-deficient HFHSD induced the progression of HS, including hepatic fibrosis (HFHSD-SH). The 1% cholesterol and 0.1% deoxycholic acid-containing HFHSD induced hypercholesterolemia (HFHSD-Chol). It is of note that these three models led to endothelial dysfunction even without PVAT, indicating increased superoxide production in the aorta itself. The second is that the HFHSD model caused endothelial dysfunction only in the aorta with PVAT. In this case, the local superoxide production in PVAT from outside of the vessel may induce endothelial dysfunction (Figure 5).

Reactive oxygen species (ROS) are important determinants of vascular function [20]. An imbalance between ROS and the antioxidant defense system is the cause of endothelial dysfunction. Several pathological conditions, including hyperglycemia, hyperlipidemia, hypertension, and insulin resistance, can promote endothelial function with impaired NO bioavailability. NO released from endothelium plays an important role in vascular complications in T2DM. It is well known that hyperglycemia can produce ROS in endothelial cells and impair NO bioavailability [21,22]. One of the important mechanisms is that the pro-oxidative state from hyperglycemia induces the uncoupling of eNOS, resulting in the production of superoxide, instead of NO, in endothelial cells [23,24,25]. In this study, the HFHSD-DM group had significant hyperglycemia, which could increase superoxide production in the aorta, leading to impaired EDR.

NAFLD is not only associated with liver-related morbidity and mortality but also with an increased risk of CVDs because NAFLD can induce systemic inflammation, hepatic insulin resistance, oxidative stress, altered lipid metabolism, and endothelial dysfunction. As such, the relationship between NAFLD and CVDs is of great interest in terms of the prevention of cardiovascular events [26]. Indeed, several studies have shown that patients with NAFLD are at increased risk of CVD [27,28,29,30]. Moreover, endothelial dysfunction is associated with steatosis grade in patients with NAFLD [30]. It was also reported that patients with NAFLD revealed eNOS dysfunction in the plates and liver [31]. These reports indicate that NAFLD independently contributes to CVDs with endothelial dysfunction. We employed the choline-deficient HFHSD containing methionine, and mice developed steatohepatitis and hepatic fibrosis without a severe loss of body weight, consistent with previous research [32,33]. In this study, the HFHSD-SH group showed the progression of HS accompanied by hepatic fibrosis and inflammation. The superoxide production in the aorta was increased in the HFHSD-SH group. Against this background, we believe that impaired EDR in the HFHSD-SH group was caused by superoxide production due to worsened steatohepatitis.

In several reports, the endothelial dysfunction of the aorta was induced in mice fed a cholesterol-containing diet [34,35,36,37,38]. Hypercholesterolemia is closely associated with endothelial injury and dysfunction [39]. We employed a 1% cholesterol and 0.1% deoxycholic acid-containing HFHSD to make a model of obesity with hypercholesterolemia. In this study, the HFHSD-Chol group had the highest cholesterol levels and revealed marked steatohepatitis. One of the mechanisms that induced endothelial dysfunction via hypercholesterolemia was that the elevations of LDL directly increased the generation of ROS and decreased NO bioavailability. It is well known that increased superoxide production has been demonstrated in vessels from hypercholesterolemic rabbits [40]. In this study, the HFHSD-Chol group also showed increased superoxide production in the aorta and impaired EDR.

A high-fat, high-sugar diet shifts the energy balance in the body toward energy storage and alters the expression of the genes involved in glucose and lipid metabolism in the whole body. Adipose tissue plays an essential role in energy storage and supply and is an important target organ in obesity-based Mets. In recent years, it has become clear that PVAT plays an important role in the regulation of vascular function in physiological and pathological states. Most arteries are covered with PVAT, and PVAT is abundant around the aorta. Enlarged PVAT has been reported to be associated with atherosclerotic plaque development and vascular calcifications [41]. The impaired adipose tissue in certain PVAT generates ROS and inflammatory adipokines that deteriorate NO bioactivity via eNOS uncoupling in PVAT [42]. In this study, the weights of adipose tissue, including GWAT, SWAT, and PVAT, were increased in all of the HFHSD groups. Additionally, adiponectin, which plays a protective role against endothelial dysfunction and arteriosclerosis, was decreased, and inflammatory adipokines, such as leptin and TNF-α, were increased in the HFHSD-only groups. Therefore, we assessed the superoxide production of aortas both with and without PVAT. Interestingly, the HFHSD groups, excluding the HFHSD-only group, had increased superoxide production in the aortas, and all of the groups, including the HFHSD-only group, had increased superoxide production from PVAT. In the ND group, the EDR with PVAT was increased compared with the EDR without PVAT. In contrast, in all of the HFHSD groups, the EDR was decreased if PVAT was preserved. These data revealed that PVAT in normal conditions could enhance the EDR by releasing relaxation factors or increasing NO bioactivity; conversely, PVAT in the HFHSD might release contraction factors and impair NO bioactivity with inflammation, leading to endothelial dysfunction. Our findings showed that endothelial dysfunction in obesity was aggravated by aortic oxidative stress from hyperglycemia, steatohepatitis, and hypercholesterolemia. However, oxidative stress in PVAT was induced with insulin resistance, which preceded the aortic endothelial dysfunction in mice models of Mets. Similar to current study, PVAT intervention in eNOS uncoupling has been investigated with some drugs (via AKT and AMPK) and exercise [12].

This study has some limitations. It was conducted in mice, and caution should be exercised in interpreting the results as directly applicable to humans. The HFHSD-DM in this study is STZ-induced DM and not a perfect model of obesity-induced DM. This may have resulted in different EDR phenotypes.

## 4. Materials and Methods

### 4.1. Animals and Diets

Male C57BL/6 mice aged less than 8 weeks were obtained from Oriental Yeast (Tokyo, Japan). They were fed a normal diet (ND; CLEA Japan, Inc., Tokyo, Japan) (Appendix A). The animals were maintained in a temperature-controlled facility on a 12 h light/12 h dark cycle (lights on from 7:00 am to 7:00 pm). Eight-week-old mice were fed with an ND, HFHSD (F2HFHSD with 28.3% of calories from carbohydrates, 54.5% from fat, and 17.2% from protein (Oriental yeast)) (Appendix A), steatohepatitis choline-deficient HFHSD (HFHSD-SH), and HFHSD containing 1% cholesterol and 0.1% deoxycholic acid (HFHSD-Chol) for 16 weeks. At 20 weeks old, some HFHSD-fed mice received streptozocin (STZ) (50 mg/kg/day for 2 days) to render T2DM (HFHSD-DM). Their body weights were measured every 2 weeks.

All experiments were performed according to the institutional ethical guidelines for animal experiments for gene manipulation experiments of the National Defense Medical College. This study was approved by the Committee for Animal Research of the National Defense Medical College.

### 4.2. Measurement of Glucose and Lipid Metabolism

At 24 weeks old, after fasting for 16 h, blood glucose levels were measured using a glucose-detection kit (439-90901, Wako Pure Chemical Industries, Osaka, Japan). Serum insulin levels were measured using an enzyme-linked immunosorbent assay (ELISA) kit (634-01481 Wako Pure Chemical Industries, Osaka, Japan).

Serum triglycerides (TGs), cholesterol, high-density lipoprotein cholesterol (HDL-C), free fatty acid (FFA), and alanine aminotransferase (ALT) were measured using enzymatic assays (432-40201, 439-17501, 431-52501, 279-75401, and 431-30901, Wako Pure Chemical Industries, Osaka, Japan).

An oral glucose tolerance test (OGTT) was conducted after fasting the mice for 16 h. After oral administration of D-glucose (1.5 g/kg), we collected blood samples at 0, 15, 30, 60, and 120 min. An insulin tolerance test (ITT) was conducted after fasting the mice for 4 h. After the injection of insulin (0.50 U/kg body weight) (Humulin R; Eli Lilly and Company, Indianapolis, IN, USA), we collected blood samples at 0, 15, 30, 45, 60, and 120 min to evaluate the insulin response. Serum blood glucose levels were measured using a glucometer (Accu-Check, Roche Japan, Tokyo, Japan). Serum adiponectin and leptin were measured using ELISA kits (Wako Shibayagi Co., Ltd., Gunma, Japan). Serum tumor necrosis factor-alpha (TNF-α) was measured using ELISA kits (R&D Systems, Minneapolis, MN, USA).

### 4.3. Tissue Preparation and Histology

Mice were sacrificed under isoflurane inhalation anesthesia (4–5%), and adequate anesthesia was confirmed by the disappearance of the paw reflex. After exposing the thoracic cavity through median sternotomy, blood was collected via puncture of the apex of the heart, and organs were perfused with 0.9% saline. The livers were fixed in 4% paraformaldehyde for 24 h, embedded in paraffin, and sectioned in 3 µm slices. The livers were stained with hematoxylin and eosin, and Masson’s trichrome. The lipid drops of livers were assessed with oil red O stain (O0625, Sigma-Aldrich, Tokyo, Japan). The frozen sections of the livers were cut into 10 µm thick sections and placed on glass slides. The samples were incubated at 37 °C for 15 min with oil red O.

All images were observed using a microscopic system (BZ-X710, Keyence, Osaka, Japan).

### 4.4. Hepatic TC and TG Measurement

Livers were perfused and homogenized in a homogenization buffer at a concentration of 5 mL/g liver tissue. Hepatic TG and cholesterol were measured using enzymatic assays (432-40201, 439-17501, Wako Pure Chemical Industries, Osaka, Japan).

### 4.5. Measurement of Systolic Blood Pressure and Heart Rates

Systolic blood pressure and heart rates were measured in a dark place using the tail-cuff method (MK-2000; Muromachi Kikai, Tokyo, Japan) without anesthesia.

### 4.6. Measurement of Vascular Superoxide Production

The superoxide production of aortic rings and PVAT were assessed with dihydroethidium (DHE) (D11347, Invitrogen, MA, USA) as previously described [20]. The frozen sections of aortic rings both without and with PVAT were cut into 10 µm thick sections and placed on glass slides. The samples were incubated at room temperature for 30 min with DHE (2 × 10^−6^ mol/L) and protected from light. Images of the samples were observed using a microscopic system (BZ-X710, Keyence, Osaka, Japan) with an excitation wavelength of 540 nm and an emission wavelength of 605 nm. The fluorescence intensity of the DHE staining was measured using a BZX analyzer (Keyence, Osaka, Japan). In addition, we employed the tempol (176141, Sigma-Aldrich, MO, USA), a superoxide scavenger, in order to test the contribution of superoxide production for the aorta and PVAT. We incubated the aortic rings both without and with PVAT, with tempol (10^−4^ mol/L) for 1 h, and then made the slides of these samples with tempol. The samples were incubated at room temperature for 30 min with DHE (2 × 10^−6^ mol/L) and protected from light. Images of the samples were observed using a microscopic system (BZ-X710, Keyence, Osaka, Japan) with an excitation wavelength of 540 nm and an emission wavelength of 605 nm. The fluorescence intensity of the DHE staining was measured using a BZX analyzer (Keyence, Osaka, Japan).

### 4.7. Isometric Tension Measurement with Organ Chamber Experiments

The assessment of isometric tension was performed as described previously [43]. Endothelial-dependent relaxation (EDR) was assessed in the thoracic aorta of mice with or without PVAT. To test the vascular tone without PVAT, PVAT was carefully removed to preserve the endothelium and cut into 3 mm rings. To test the vascular tone with PVAT, the thoracic aorta with PVAT was cut into 3 mm rings. The aortas with or without PVAT were mounted in organ baths filled with a Krebs–Ringer bicarbonate solution (NaCl 118.3 mmol/L, KCl 4.7 mmol/L, CaCl_2_ 2.5 mmol/L, MgSO_4_ 1.2 mmol/L, KH_2_PO_4_ 1.2 mmol/L, NaHCO_3_ 25 mmol/L, D-glucose 5.5 mmol/L) aerated with 95% O_2_ and 5% CO_2_ at 37 ℃. The aortas with or without PVAT were attached to a force transducer Easy Magnus UC-5A (Iwashiya Kishimoto Medical Instruments, Kyoto, Japan), and isometric tensions were recorded. Pressure of 0.25 g was applied to all aortic rings with and without PVAT for 10 minutes, increasing by 0.25 g every 10 minutes until 1 g was reached. All aortic rings were immersed in a KCl solution (60 mmol/L) and precontracted with L-phenylephrine (MP Biomedicals) (10^−6^ mol/L) for submaximal contraction. All aortic rings were exposed to increasing concentrations of acetylcholine (A6625, Sigma-Aldrich, Tokyo, Japan) from 10^−9^ to 10^−5^ mol/L after the contraction curve reached a plateau. To test the contribution of superoxide production for endothelial dysfunction, we employed tempol. The same aortic rings with or without PVAT were pretreated with tempol (10^−4^ mol/L) for 1 h before precontraction with L-phenylephrine.

### 4.8. Statistical Analysis

The results are presented as the mean ± standard deviation (SD) or standard error of the mean (SEM). Data for body weights, liver weights, fat weights (gonadal white adipose tissue (GWAT), subcutaneous white adipose tissue (SWAT)), PVAT, systolic blood pressure, heart rates, and serum parameters were evaluated using one-way analysis of variance (ANOVA), followed by a post hoc test with Bonferroni’s correction for multiple comparisons. OGTT, ITT, and vascular relaxation were evaluated using two-way ANOVA with repeated measures, followed by a post hoc test with Bonferroni’s correction for multiple comparisons. All data were analyzed with GraphPad Prism Software version 7.03 (GraphPad Software, La Jolla, CA, USA). *p*-value was represented by * *p* < 0.05, ** *p* < 0.01, and *** *p* < 0.001 versus the ND group and † *p* < 0.05, †† *p* < 0.01, and ††† *p* < 0.001 versus the HFHSD group.

## 5. Conclusions

HFHSD alone did not cause endothelial dysfunction in the aortas without PVAT. However, additional factors, such as high serum glucose, steatohepatitis, and high serum cholesterol, may cause impaired EDR without PVAT. The EDR with PVAT was impaired by the HFHSD alone. Impaired EDR is caused by increased superoxide production from the aorta and PVAT.

## Figures and Tables

**Figure 1 ijms-24-06486-f001:**
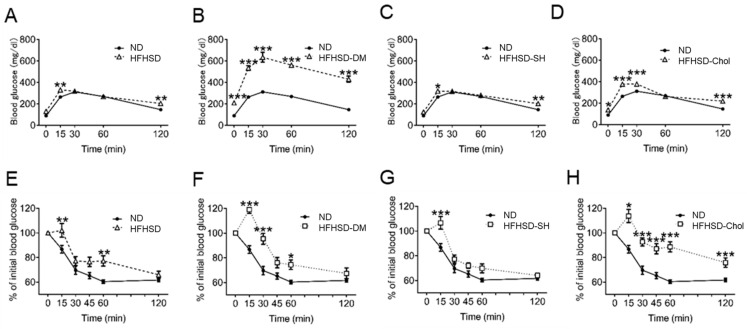
All the HFHSD groups revealed higher glucose levels and insulin resistance, and the HFHSD-DM group, in particular, showed the highest levels of glucose: (**A**–**D**) Oral glucose tolerance test (OGTT) time curve for glucose levels (n = 8); (**E**–**H**) insulin tolerance test (ITT) time curve for glucose levels (n = 8). OGTT and ITT are expressed as mean blood glucose levels ± standard error of the mean (SEM). ND, normal diet group; HFHSD, high-fat/high-sucrose (HFHSD) diet group; HFHSD-DM, HFHSD group that received streptozocin (STZ) (50 mg/kg/day for 2 days at 20 weeks old); HFHSD-SH, choline-deficient HFHSD group; HFHSD-Chol, 1% cholesterol and 0.1% deoxycholic acid-containing HFHSD group. * *p* < 0.05, ** *p* < 0.01, and *** *p* < 0.001 compared with the ND group.

**Figure 2 ijms-24-06486-f002:**
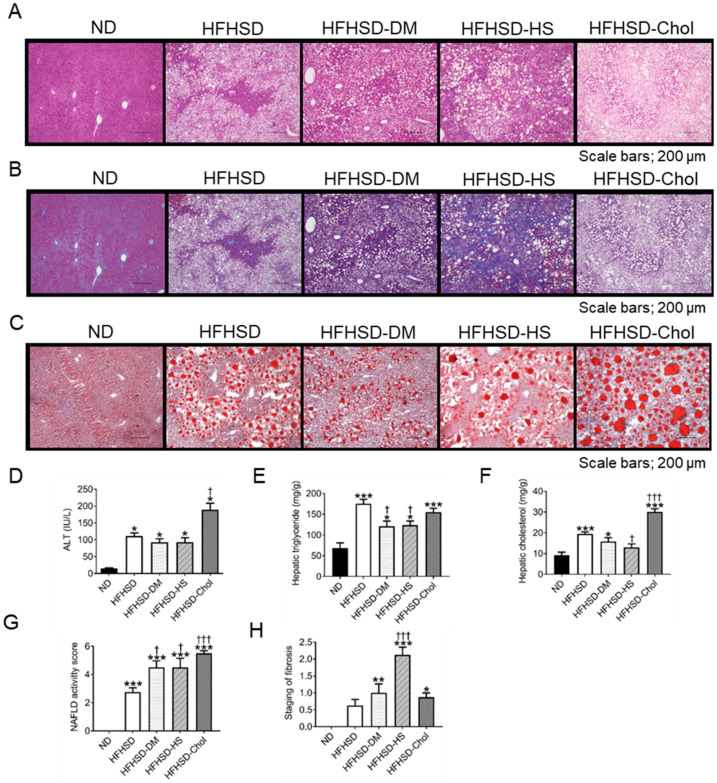
HFHSD-SH had the worst steatohepatitis and HFHSD-Chol had the worst steatohepatitis: (**A**) Hematoxylin and eosin staining of the liver for the ND, HFHSD, HFHSD-DM, HFHSD-SH, and HFHSD-Chol. Scale bars, 200 µm; (**B**) Masson’s trichrome staining of the liver for the ND, HFHSD, HFHSD-DM, HFHSD-SH, and HFHSD-Chol. Scale bars, 200 µm; (**C**) oil red O staining of the liver for the ND, HFHSD, HFHSD-DM, HFHSD-SH, and HFHSD-Chol. Scale bars, 200 µm; (**D**) serum ALT levels (n = 12–16); (**E**,**F**) hepatic cholesterol and triglyceride content (n = 8); (**G**,**H**) non-alcoholic fatty liver disease (NAFLD) activity score (n = 8) and staging of fibrosis (n = 8) were adapted from Brunt et al. ND, normal diet group; HFHSD, high-fat/high-sucrose (HFHSD) diet group; HFHSD-DM, HFHSD group that received streptozocin (STZ) (50 mg/kg/day for 2 days at 20 weeks old); HFHSD-SH, choline-deficient HFHSD group; HFHSD-Chol, 1% cholesterol and 0.1% deoxycholic acid-containing HFHSD group; ALT, alanine aminotransferase; NAFLD, non-alcoholic fatty liver disease. Error bars represent the standard error of the mean (SEM). * *p* < 0.05, ** *p* < 0.01, and *** *p* < 0.001 compared with the ND group and † *p* < 0.05 and ††† *p* < 0.001 compared with the HFHSD group.

**Figure 3 ijms-24-06486-f003:**
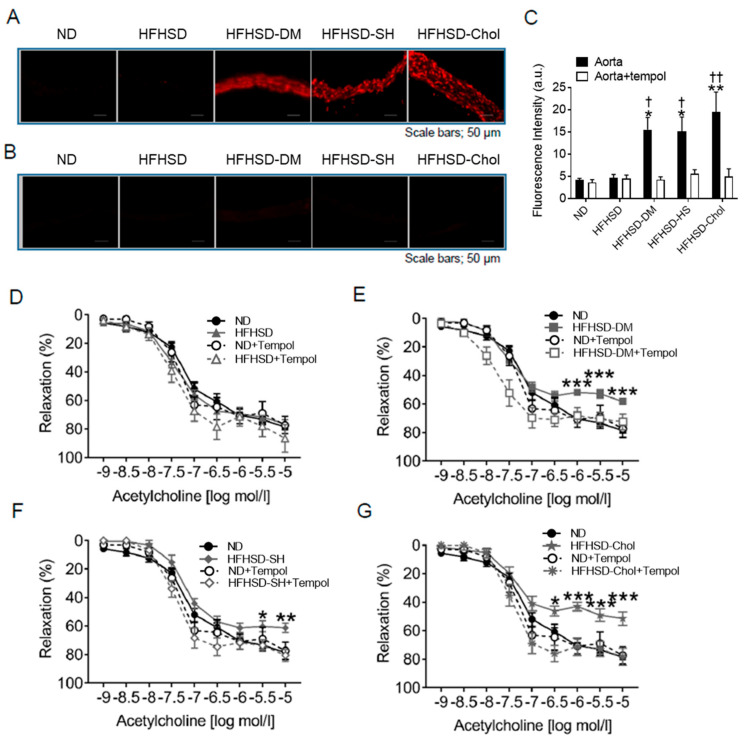
Endothelial-dependent relaxation in the HFHSD-DM, HFHSD-SH, and HFHSD-Chol groups was significantly impaired and accompanied by increased superoxide production from aortas without PVAT: (**A**) DHE staining of aortic rings and (**B**) aortic rings with tempol; (**C**) quantification of DHE fluorescence (n = 8 for each group). Vascular relaxation of aortic rings with acetylcholine in the HFHSD group (**D**), HFHSD-DM group (**E**), HFHSD-SH group (**F**), and HFHSD-Chol group (**G**) compared with the ND group (ND, n = 16; HFHSD, n = 12; HFHSD-DM, n = 12; HFHSD-SH, n = 16; HFHSD-Chol, n = 14). Acetylcholine indicates acetylcholine; DHE, dihydroethidium; ND, normal diet group; HFHSD, high-fat/high-sucrose (HFHSD) diet group; HFHSD-DM, HFHSD group that received streptozocin (STZ) (50 mg/kg/day for 2 days at 20 weeks old); HFHSD-SH, choline-deficient HFHSD group; HFHSD-Chol, 1% cholesterol, and 0.1% deoxycholic acid-containing HFHSD group. Error bars represent the standard error of the mean (SEM). * *p* < 0.05, ** *p* < 0.01, and *** *p* < 0.001 compared with the ND group and † *p* < 0.05 and †† *p* < 0.01 compared with aorta without treatment of tempol.

**Figure 4 ijms-24-06486-f004:**
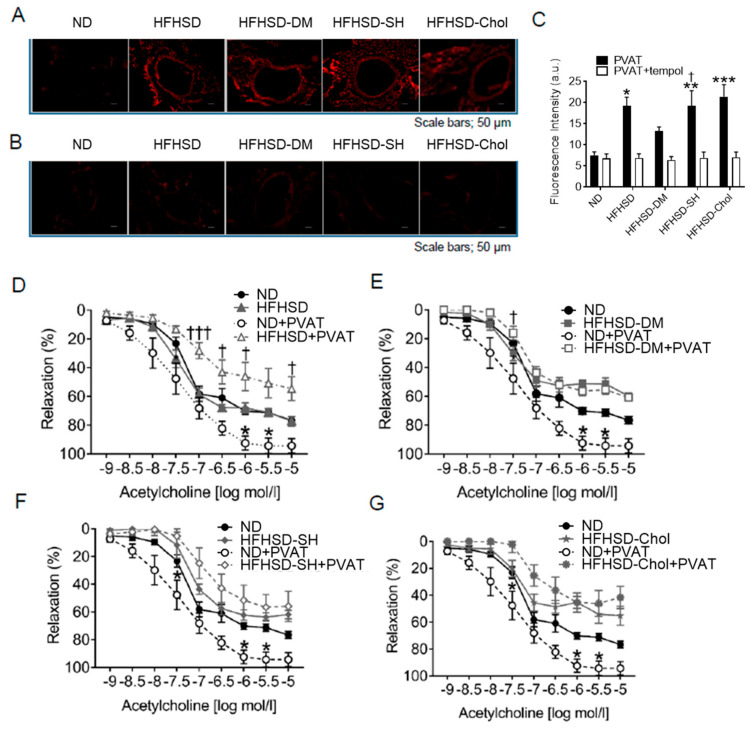
PVAT increased EDR in the ND group but decreased EDR in the HFHSD group and was accompanied by increased superoxide production from PVAT: (**A**) DHE staining of aortic rings with PVAT and (**B**) with tempol; (**C**) quantification of DHE fluorescence (n = 8 for each group). Vascular relaxation of aortic rings with PVAT or without PVAT on acetylcholine in the HFHSD group (**D**), HFHSD-DM group (**E**), HFHSD-SH group (**F**), and HFHSD-Chol group (**G**) (n = 8 for each group). Acetylcholine indicates acetylcholine; PVAT, perivascular adipose tissue; DHE, dihydroethidium; ND, normal diet group; HFHSD, high-fat/high-sucrose (HFHSD) diet group; HFHSD-DM, HFHSD group that received streptozocin (STZ) (50 mg/kg/day for 2 days at 20 weeks old); HFHSD-SH, choline-deficient HFHSD group; HFHSD-Chol, 1% cholesterol and 0.1% deoxycholic acid-containing HFHSD group. Error bars represent the standard error of the mean (SEM). * *p* < 0.05, ** *p* < 0.01, and *** *p* < 0.001 compared with the ND group and † *p* < 0.05 and ††† *p* < 0.001 compared with the HFHSD group.

**Figure 5 ijms-24-06486-f005:**
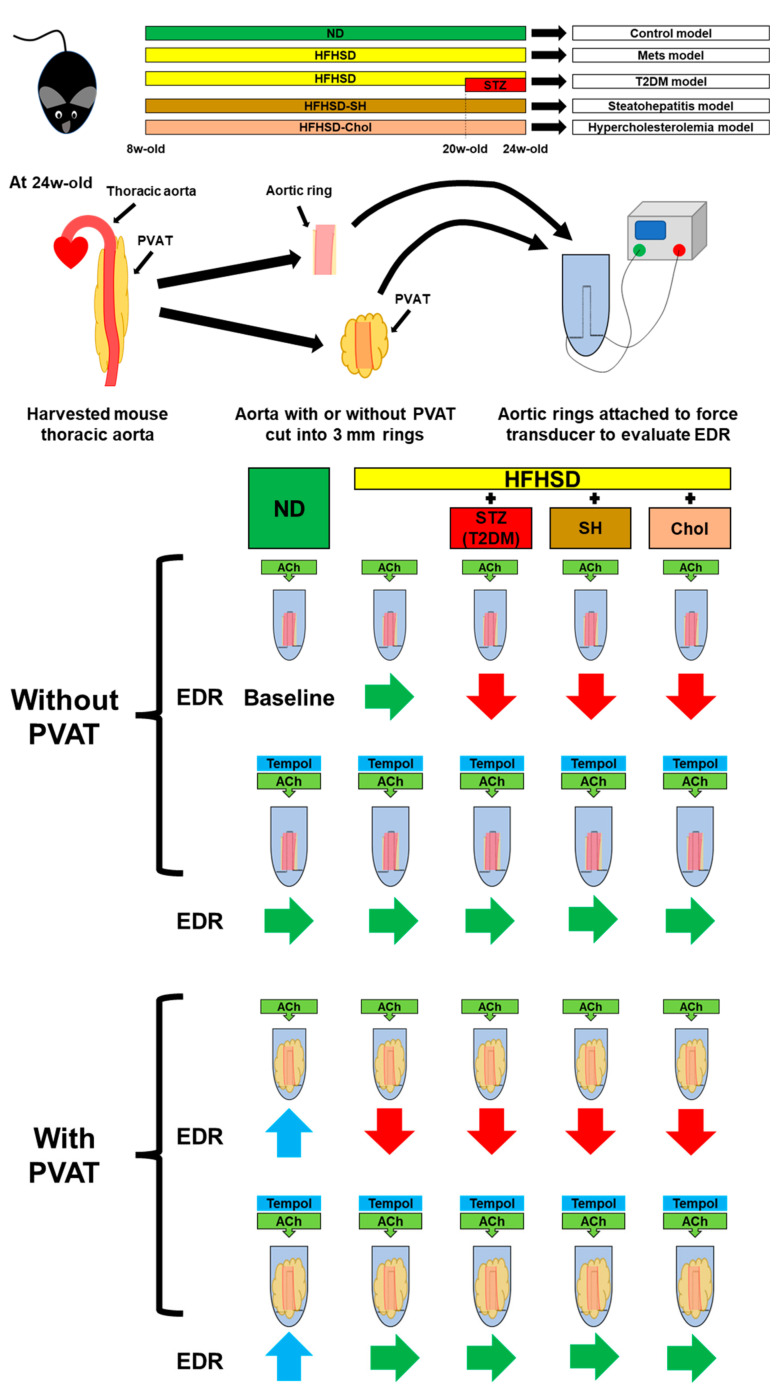
Summary of experimental results. ACh, acetylcholine; EDR, endothelium-dependent relaxation; PVAT, perivascular adipose tissue; STZ, streptozocin; T2DM, type 2 diabetes mellitus; ND, normal diet group; HFHSD, high-fat/high-sucrose (HFHSD) diet group; HFHSD-DM, HFHSD group that received STZ (50 mg/kg/day for 2 days at 20 weeks old); HFHSD-SH, choline-deficient HFHSD group; HFHSD-Chol, 1% cholesterol and 0.1% deoxycholic acid-containing HFHSD group.

**Table 1 ijms-24-06486-t001:** Body, liver, and adipose tissue weight.

	ND	HFHSD	HFHSD-DM	HFHSD-HS	HFHSD-Chol
Body weight (g)	27.5 ± 0.47	43.9 ± 0.64 *	38.1 ± 0.99 *	41.2 ± 0.79 *	42.0 ± 2.13 *
Liver weight (mg)	104.1 ± 7.3	230.4 ± 14.9 *	190.3 ± 13.2 *	183.5 ± 12.4 *	319.9 ± 12.1 *^†^
GWAT weight (mg)	59.6 ± 4.0	165.5 ± 5.1 *	182.0 ± 5.8 *	188.8 ± 8.0 *	195.4 ± 7.4 *^†^
SWAT weight (mg)	37.1 ± 4.6	216.3 ± 12.6 *	177.1 ± 22.0 *^†^	167.1 ± 18.9 *^†^	270.3 ± 11.0 *
PVAT weight (mg)	3.0 ± 0.27	11.0 ± 0.81 *	7.2 ± 0.51 *^†^	7.8 ± 0.68 *^†^	12.6 ± 0.62 *

Data are presented as mean ± SE with n = 12–16 per group in body, liver and GWAT. n = 8 per group in SWAT and PVAT, * *p* < 0.05 compared to ND and † *p* < 0.05 compared to HFHSD. ND indicates normal diet group; HFHSD, high-fat/high-sucrose (HFHSD) diet group; HFHSD-DM, HFHSD group, which were received streptozocin (STZ) (50 mg/kg/day for 2 days on the age of 20 weeks); HFHSD-HS, choline deficient HFHSD group; HFHSD-Chol, 1% cholesterol and 0.1% deoxycholic acid containing HFHSD group; GWAT, gonadal white adipose tissue; SWAT, subcutaneous white adipose tissue; PVAT, perivascular adipose tissue; BP, blood pressure.

**Table 2 ijms-24-06486-t002:** Serum glucose and lipid metabolic parameters and adipokines.

	ND	HFHSD	HFHSD-DM	HFHSD-HS	HFHSD-Chol
**Glucose metabolic parameters**					
Liver weight (mg)	104.1 ± 7.3	230.4 ± 14.9 *	190.3 ± 13.2 *	183.5 ± 12.4 *	319.9 ± 12.1 *^†^
GWAT weight (mg)	59.6 ± 4.0	165.5 ± 5.1 *	182.0 ± 5.8 *	188.8 ± 8.0 *	195.4 ± 7.4 *^†^
SWAT weight (mg)					
PVAT weight (mg)	3.0 ± 0.27	11.0 ± 0.81 *	7.2 ± 0.51 *^†^	7.8 ± 0.68 *^†^	12.6 ± 0.62 *
**Lipid metabolic parameters**					
Triglyceride (mg/dL)	79.69 ± 6.89	69.48 ± 2.33	58.12 ± 6.50	82.35 ± 14.26	63.67 ± 5.35
Cholesterol (mg/dL)	90.82 ± 3.59	191.5 ± 11.96 *	152.2 ± 10.88 *	152.7 ± 13.48 *	258.5 ± 12.68 *^✝^
HDL-cholesterol (mg/dL)	71.56 ± 4.18	145.9 ± 13.63 *	105.4 ± 7.459	129.1 ± 15.78 *	149.0 ± 4.77 *
Free fatty acid (mEq/L)	1.39 ± 0.11	1.24 ± 0.11	1.05 ± 0.075	1.35 ± 0.079	0.98 ± 0.050 *
**adipokines**					
Adiponectin (ng/mL)	2617 ± 323.9	1284 ± 116.6 *	1202 ± 77.23 *	1081 ± 93.59 *	1026 ± 88.18 *
Leptin (pg/mL)	7.31 ± 2.63	52.23 ± 2.41 *	28.76 ± 3.93 *^✝^	31.83 ± 3.73 *	44.79 ± 2.34 *
TNF-α (pg/mL)	1.63 ± 0.49	6.78 ± 0.48 *	5.48 ± 1.02 *	5.49 ± 0.43 *	9.00 ± 1.42 *

Fasting glucose with n = 8 per group, fasting insulin with n = 10–14 per group, Triglyceride, cholesterol, HDL-cholesterol, free fatty acid with n = 12–16 per group, Adiponectin with n = 8 per group, Leptin with n = 7–8 per group, TNF-α with n = 9–10 per group. Data are presented as mean ± SE. * *p* < 0.05 compared to ND and † *p* < 0.05 compared to HFHSD. ND indicates normal diet group; HFHSD, high-fat/high-sucrose (HFHSD) diet group; HFHSD-DM, HFHSD group, which were received streptozocin (STZ) (50 mg/kg/day for 2 days on the age of 20 weeks); HFHSD-HS, choline deficient HFHSD group; HFHSD-Chol, 1% cholesterol and 0.1% deoxycholic acid containing HFHSD group; HDL, high-density lipoprotein; TNF, tumor necrosis factor.

## Data Availability

Data supporting the reported results can be obtained from the corresponding author upon reasonable request.

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
