# Peer review of "Reactive Oxygen Species in the Aorta and Perivascular Adipose Tissue Precedes Endothelial Dysfunction in the Aorta of Mice with a High-Fat High-Sucrose Diet and Additional Factors"

_ijms, 2023, doi:10.3390/ijms24076486_

Round 1
Reviewer 1 Report
In this manuscript, the authors have found that reactive oxygen species in perivascular adipose tissue preceded to endothelial dysfunction in aorta from mice with high fat high sucrose diet. Following my careful review, I can find that this study is well-conducted and organized, and the used experimental groups are appropriate. The experimental methods and analyzed parameters are suitable for this kind of studies. The results are appropriately described, and the figures are in a good order and well-prepared. The discussion is well written and explains the findings in a good order, and the conclusion is supported by the results. Therefore, I recommend this manuscript for publication following some corrections.
1- Generally, the abstract should be rewritten. It is not informative and does not reflect the aim of the research and the obtained results. It is preferable to make the abstract one paragraph not segmented.
2- In the abstract you should add p value when you mention the word significantly.
3- The introduction not sufficient, some of the little flaws include, the lack of organization in presented information, while there are also some statements are without corresponding citation. please add more recent references in this regard.
4- The authors have to incorporate previous literature about the basic causes of endothelial dysfunction and what is the novel in this study.
5- Abbreviations should be fully described in the first mention and not repeated every time.
6- It is important to clearly state the rationale and aim of the study and why the study focused on perivascular adipose tissue?? .
7- The total number of animals utilized in the study, as well as the numbers employed in each group, should be clearly mentioned by the authors and in the parameters being examined.
8- The authors are required to clarify how they scarified the animals and method of blood and tissues collection as well as type of anesthesia ?
9- The exact source, concentrations and the catalogue numbers of the used kits and chemicals should be mentioned.
10- A collective figure that summarizes the results is recommended.
11- I think that it is preferrable to express the results as mean ± SD.
12- The manuscript should be revised by English-naïve speaker to improve the quality of the language. The manuscript should be checked regarding the grammatical errors and plagiarism.
Reviewer 2 Report
The authors aimed to determine the additional interventions for the induction of endothelial dysfunction in the aorta from mice with high fat and high sucrose diet (HFHSD). They found that endothelial-dependent relaxation (EDR) in HFHSD-DM, HFHSD-HS, and HFHSD-Chol groups were significantly impaired accompanied by the elevation of vascular superoxide production. Tempol, a superoxide scavenger, restored EDR in HFHSD-DM, HFHSD-HS, and HFHSD-Chol. Perivascular adipose tissue (PVAT) increased EDR in ND group, whereas decreased EDR in HFHSD group was accompanied by increased superoxide production from PVAT. Tempol normalized EDR in the aorta with PVAT in HFHSD group. The authors concluded that HFHSD alone did not cause endothelial dysfunction in the aorta without PVAT. However, additional factors, such as hyperglycemia, hepatosteatitis, and hypercholesterolemia, caused the impaired EDR without PVAT. Moreover, EDR with PVAT was impaired by HFHSD alone accompanied by the increased superoxide production from PVAT.
This is a well-organized, well-conducted and nicely documented study.
Comments
1. Table 1 and 2. Abbreviations if Table 1 and 2 should be explained in the footnote of the table.
2. Some possible clinical consequences of the findings could be discussed. Why is it important that PVAT can induce EDT without any additional factors? How could we decrease superoxide production? Are there any possible therapeutic consequences?
3. Some limitations of the study could be mentioned. Results of animal studies are not always in line with the results of human studies. Although STZ-induced DM is a widely accepted model of T2DM, there are several phenotypic differences between STD-induced and obesity-induced DM.
4. There are some typos throughout the manuscript. Ln 109: “Livers” instead of “livers”, extra space in Ln 403, etc.
5. Use of steatohepatitis instead of hepatosteatitis would be better.
6. There are several not widely used abbreviations. A list of abbreviations could be useful.
7. English need some editing. Please, rephrase the first sentence of the Discussion.
8. Running head should be removed.
Round 2
Reviewer 1 Report
The authors have answered all of my questions and the paper has been greatly improved. Therefore, it can be accepted for publication.